# RETRACTED: Synthesis of New Magnetic Crosslinked Poly (Ionic Liquid) Nanocomposites for Fast Congo Red Removal from Industrial Wastewater

**DOI:** 10.3390/nano9091286

**Published:** 2019-09-09

**Authors:** Ayman M. Atta, Abdelrahman O. Ezzat, Yaser M. Moustafa, Nourah I. Sabeela, Ahmed M. Tawfeek, Hamad A. Al-Lohedan, Ahmed I. Hashem

**Affiliations:** 1Chemistry Department, College of Science, King Saud University, Riyadh 11451, Saudi Arabia; ao_ezzat@yahoo.com (A.O.E.); noonah-37@hotmail.com (N.I.S.); atawfik@ksu.edu.sa (A.M.T.); hlohedan@ksu.edu.sa (H.A.A.-L.); 2Egyptian Petroleum Research Institute, Nasr City 1772, Cairo, Egypt; ymoustafa12@yahoo.com; 3Chemistry Department, Faculty of Science, Ain Shams University, Abasia 11566, Cairo, Egypt; emyhashem2004@yahoo.com

**Keywords:** magnetite, cross-linking, poly(ionic liquids), 4-vinylpyridine-co-acrylamide, adsorbents

## Abstract

Advanced materials reliant on cross-linked magnetic poly (ionic liquids) (PILs) have been widely utilized in environmental applications for water purification. The present work demonstrates our preparation of a new magnetic cross-linked PIL based on quaternized 4-vinyl-pyridine-co-acrylamide (QVP/AAm). The chemical composition, thermal stability, magnetic properties, morphology, particle sizes, and zeta potential of the magnetic QVP/AAm composites were investigated. Fast adsorption and desorption kinetics, high adsorption capacity, rapid magnetic separation, and the absence of secondary pollution in the adsorption process make QVP/AAm-Fe_3_O_4_ a highly effective adsorbent for the elimination of anionic acidic Congo red contaminants from industrial wastewater.

## 1. Introduction

Organic dyes present in wastewater from the textile, paper, leather, food, cosmetic, and pharmaceutical industries pollute larger water bodies and thus affect the ecological system and potentially harm human health [1,2,3]. Among the various treatment options, which include chemical oxidation, photo-catalytic degradation, biological treatment, membrane filtration and ion-exchange techniques, the adsorption of organic dyes from polluted waters has attracted considerable interest as a practical way to purify industrial wastewater [4,5,6,7]. The adsorption technique has various advantages—it is simple, highly efficient, and generates no secondary pollutants. Magnetic and porous materials have been proposed as efficient environment-friendly adsorbents owing to their ability to clean up organic and inorganic water pollutants with high speed and efficiency and their easy manipulation using an external magnet [8,9,10]. In general, the agglomeration of magnetic nanoparticles reduces their efficiency as effective adsorbents owing to the weakening of their magnetic properties and blocking of their active sites. Ionic liquids (ILs) remove water pollutants effectively and are recommended for application as green adsorbents owing to their unique chemical composition, thermal stability, and excellent designability [11,12,13]. Nevertheless, several drawbacks restrict the application of ILs as adsorbents for wastewater treatment, such as the tedious separation procedures and difficult recyclability. Recently, magnetic ILs has been prepared as adsorbents but they have a few limitations, such as poor stability, unequal distribution, and small load volumes [14,15,16]. 

Hyperbranched poly (ionic liquid) (PIL) adsorbents with imidazolium backbones are capable of highly efficient anionic dye removal from industrial wastewater, but they have low recycling potentials [17]. Cross-linked ILs, modified with nanoporous magnetic core/shell nanocomposites and multi-layer functionalized PIL coated magnetic nanoparticles, have been suggested as effective and highly adsorbent catalysts to remove water pollutants several times [18,19,20]. The surface of the Fe_3_O_4_ nanoparticles can be modified with active vinyl groups such as 3-methacryloxypropyltrimethoxy to polymerize with PIL networks; these nanoparticles can be easily recovered and recycled by using an external magnet [20]. The extraction performance of magnetic polymer nanocomposites can be easily modified to achieve higher extraction capacity in the presence of PILs as surface modifiers for nanoparticles [21]. Vinylpyridine polymer (VP) PIL nanocomposites have a unique potential in the extraction field which makes their synthesis and application as adsorbents increasingly popular [22,23,24,25]. The quaternization of VP after polymerization has been discussed previously to produce polyelectrolyte or hydrogel adsorbents [26,27]. In this work, novel cationic quaternized 4-vinylpyridine (QVP) functional with dichloro diethyl ether is copolymerized and cross-linked with acrylamide (AAm) to produce QVP/AAm PIL. Its magnetic nanocomposite is prepared based on Fe_3_O_4_ coated with QVP/AAm to form new porous magnetic PIL composites as adsorbents for the removal of organic dyes from industrial wastewater. The structure and adsorption characteristics for the removal of anionic dyes of the synthesized QVP/AAm PIL magnetite composites and the pure Fe_3_O_4_ NPs have also been studied.

## 2. Experimental

### 2.1. Materials

Vinylpyridine monomer (VP), azobisisobutyronitrile (AIBN), and bis(2-chloroethyl)ether (DCDE) were purchased from Sigma Chem. Co. (St. Louis, MO, USA). Acrylamide (AAm) monomer and N,N-dimethylformamide (DMF) were purchased from Aldrich (Munich, Germany). High yield magnetite nanoparticles (Fe_3_O_4_ NPs) were prepared from the reaction of anhydrous FeCl_3_ and KI (obtained from Aldrich; Munich, Germany) after iodine removal in the presence of ammonia as reported in previous works [28]. Congo red (CR) dye purchased from Sigma-Aldrich Co. (St. Louis, MO, USA) is used to prepare stock solutions of 50–500 mg L^−1^. Phosphate buffer solution (H_3_PO_4_/NaH_2_PO_4_) in the presence of 0.1 M HCl or 0.1 M NaOH is used to adjust the pH of aqueous solution in the 2–3 or 7–12 range, respectively.

### 2.2. Preparation Methods

#### 2.2.1. Preparation of Magnetic Cross-Linked PIL 

To quaternize 4-vinylpyridine (QVP), 4-vinylpyridine (6.5 g, 0.06 mol) and dichloro diethyl ether (4.25 g, 0.03 mol) were dissolved in DMF (25 mL) and stirred under a nitrogen atmosphere at room temperature for 5 h to obtain a clear solution. After that, the temperature was raised to 323 K for another 5 h. AAm (4.25 g, 0.06 mol) was dissolved in DMF (10 mL) and added to the reaction mixture. AIBN initiator (0.1 wt% related to the weights of QVP and AAm) was injected into the stirred reaction mixture. The reaction temperature was set at 338 K for 5 h. The yellow precipitate of the cross-linked QVP/AAm was separated by filtering it out from the solution and washed three times with ethanol. QVP monomer was separated for characterization before adding the AIBN by extraction its DMF solution with diethyl ether to remove the unreacted product that separated in DMF. The diethyl ether extract dried at 313 K under high vacuum to produce a yellow slightly viscous liquid with a refractive index of 1.4741 (n 20/D). The QVP yield% is 93.2%. 

Fe_3_O_4_ NPs (0.2 g) and QVP/AAm (0.4 g) were dispersed separately in deionized water (12.5 mL) using a probe sonicator for 30 min. After complete dispersion of the two solutions, they were mixed together by continuous stirring overnight. The magnetic cross-linked PIL QVP/AAm-Fe_3_O_4_ was collected using an external magnet.

#### 2.2.2. Preparation of Magnetic Cross-Linked PIL (Fe_3_O_4_-QVP/AAm)

The procedure used to prepare QVP/AAm was repeated to prepare magnetic Fe_3_O_4_-QVP/AAm except that Fe_3_O_4_ NPs (0.03 mol) and NaOH powder (0.03 mol) were dispersed with VP (9.8 g, 0.09 mol) and DCDE (8.5 g, 0.06 mol), dissolved in DMF (50 mL), and stirred under a nitrogen atmosphere at room temperature for 5 h. AAm (6.5 g, 0.09 mol) was dissolved in DMF (15 mL) and added to the reaction mixture while raising the temperature of the reaction mixture to 323 K for another 5 h. AIBN initiator (0.1 wt% related to QVP and AAm weights) was injected into the reaction mixture under nitrogen atmosphere and the reaction temperature was adjusted to 338 K for 5 h. The black precipitate of cross-linked Fe_3_O_4_-QVP/AAm was separated and washed several times with ethanol. The product was kept in an oven at 70 °C overnight to obtain pure dried cross-linked Fe_3_O_4_-QVP/AAm.

### 2.3. Characterization

The chemical structure of QVP was investigated by ^1^H-NMR and ^13^C-NMR spectroscopy (400 MHz Bruker Avance DRX-400 spectrometer, Toronto, ON, Canada). The chemical structures of Fe_3_O_4_, Fe_3_O_4_-QVP/AAm, QVP/AAm-Fe_3_O_4_ and QVP/AAm were investigated using a Fourier transform infrared (FTIR) spectrometer (Shimadzu FTIR 8000 spectrometer, Kyoto, Japan) using KBr pellets. The size of the prepared Fe_3_O_4_, Fe_3_O_4_-QVP/AAm, QVP/AAm-Fe_3_O_4_, and QVP/AAm and their polydispersity index (PDI) in distilled water (DW) and 0.01 M of KCl were measured at 25 °C using dynamic light scattering (DLS; Zetasizer Nano ZS, Malvern Instrument Ltd., Malvern, UK). 

X-ray powder diffractometer (BDX-3300 diffractomete; Eindhoven, Netherlands) using Cu anode; *k* = 0.15406 nm, 30 kV and 10 mA used to investigated the crystalline structure of magnetic composites at 25 °C. The morphologies of the prepared materials were examined using a transmission electron microscope (TEM; JEOL JEM-2100F JEOL, Tokyo, Japan) at acceleration voltage of 200 kV) and scanning electron microscope (SEM; Nova nano, SEM 430, FEI, USA). The thermal stabilities of the synthesized magnetic composites were investigated using thermogravimetric analysis (a TGA-50 SHIMADZU, Tokyo, Japan) under nitrogen flow while increasing the temperature at the rate of 10 °C/min. A vibrating sample magnetometer (VSM; LDJ9600 in a field of 20,000 Oe, LDJ Electronics, MI, USA) was used to evaluate the magnetic properties of the materials at room temperature. Ultraviolet-visible (UV-vis) spectrophotometer (Shimadzu UV-1208 model at wavelength λmax equal to 662 nm, SHIMADZU, Tokyo, Japan) was used to determine the change in the CR dye concentration at a maximum wavelength of 496 nm. The surface area, the pore volume and the pore-size distribution of Fe_3_O_4_-QVP/AAm, QVP/AAm-Fe_3_O_4_ were estimated by the Brunauer-Emmett-Teller (BET) method using a nitrogen adsorption-desorption process. The adsorption data in the relative pressure (P/P_0_) range of 0.05–0.3 using a ASAP 2020 M apparatus (Micromeritics Instrument Corp, Norcross, GA, USA).

### 2.4. Dye Removal Measurements

A UV-visible spectrophotometer was utilized to evaluate the CR dye absorbance at a wavelength of 496 nm. CR dye at a concentration of 100 mg·L^−1^ was dispersed in 25 mL of water. Magnetic cross-linked PIL adsorbent (4 mg) was dispersed in the CR solution followed by collection using external magnetic field to study its removal with reference to different parameters. The adsorption capacities of the CR dye at equilibrium *q*_e_ (in mg·g^−1^) and the adsorption efficiency *E* (in%) were measured using the following equations:

*q*_e_ = (*C*_o_ − *C*_e_) × *V*/*m*
(1)


*E*(%) = (*C*_o_ − *C*_e_) × 100/*C*_o_
(2)

where *C*_o_, *C*_e_, *V* and *m* are the aqueous dye concentration at zero time, at equilibrium (mg L^−1^), the aqueous dye volume (L) and the mass of the adsorbent (g), respectively.

The powder was treated with 0.5 mol L^−1^ of HCl (in water/ethanol ratio of 50/50) and neutralized with 0.1 mol·L^−1^ of NaOH (in water/ethanol ratio of 50/50) solutions to remove the CR from the prepared composite. The powder was isolated from the solution by using an external magnet at the end of the experiment and reused by stirring into 25 mL of 0.1 M NaOH solution for 3 h and then washing with distilled water and drying at room temperature to be used for several adsorption experiments.

## 3. Results and Discussion

Crosslinked PIL can be obtained from the crosslinking of polymerizable IL monomers with crosslinkers containing at least two double bonds by the radical polymerization technique. In this respect, the present work prepared and purified liquid IL crosslinker QVP as reported in the experimental section. The chemical structure of the QVP elucidated from its ^1^H-NMR and ^13^C-NMR spectra is represented in Figure 1a,b, respectively. The ^1^H-NMR spectrum of QVP (Figure 1a) confirms the quaternization of 4-VP with DCDE as represented in Figure 1 by the peaks in the 7.5 ppm to 7.6–9.3 ppm region. Moreover, the comparison of the integration ratios of the CH_2_-N^+^ and vinyl double bond protons at 5–6.5 ppm due to the pyridine aromatic ring protons, is found to be 5/4 which matches the calculated ratio. The shifts for C=N and aromatic C=C rings of 4-VP from 140 and 154 ppm to 140 and 163 ppm in the ^13^C-NMR spectrum (Figure 1b) confirm the quaternization to form QVP confirms also the quaternization of 4-VP [29].

Currently, the ready preparation of cross-linked polymers from polymerizable IL monomers to produce either hydrophobic or hydrophilic cross-linked PILs suggests a significant application potential in water treatment or desalination [30]. This can be attributed to the combination of the unique properties of PILs as polyelectrolytes and ionic liquids in the fields of separation, sensing, and desalination [30]. For this purpose, 4-VP monomer was selected to prepare a polymerizable IL monomer by quaternization of the nitrogen group with a linking agent based on DCDE as represented in Figure 1 and Figure 2. The QVP can be copolymerized with AAm to form alternate, random, or block cross-linked copolymers depending on their reactivity ratios. The copolymerization of 4-VP and acrylamide monomer produced random copolymers with the formation of 4-vinylpyridine blocks [31,32,33]. The magnetite nanoparticles were used to prepare magnetic QVP/AAm nanocomposites by incorporation either during or after the cross-linking copolymerization of QVP/AAm as represented in Figure 1 and Figure 2, respectively. The magnetite NPs can interact with QVP/AAm (Figure 1) through either electrostatic interaction between the negative surface charge on the magnetite surface or hydrogen bonding with the positive charge or amide groups of the cationic QVP/AAm. It is suggested that the hydroxyl groups of the magnetite can react with DCDE in the presence of NaOH to be incorporated during the cross-linking copolymerization of QVP/AAm (Figure 2). The effects of magnetite on the chemical structure, crystallinity, particle sizes, morphologies, and thermal stabilities of Fe_3_O_4_-QVP/AAm and QVP/AAm-Fe_3_O_4_ are discussed in a forthcoming section.

### 3.1. Characterization

The chemical structures of Fe_3_O_4_, Fe_3_O_4_-QVP/AAm, QVP/AAm-Fe_3_O_4_, and QVP/AAm are confirmed by the FTIR spectra presented in Figure 2a–d. The appearance of the strong band at 591 cm^−1^ in the spectra of Fe_3_O_4_, Fe_3_O_4_-QVP/AAm, and QVP/AAm-Fe_3_O_4_ (Figure 2a–c) represents the stretching vibrations of Fe-O and indicates the presence of Fe_3_O_4_ in the polymer composites. The hydroxyl groups present on the surfaces of the Fe_3_O_4_ NPs (Figure 2a) are established from the appearance of the band at 3450 cm^−1^ related to the OH stretching vibration. Moreover, the presence of the stronger bands related to the stretching vibration of C=O (1638 cm^−1^), C−N (1516 cm^−1^), and C=C (1416 cm^−1^) indicates the incorporation of the aromatic ring of 4-VP into the polymer composites (Figure 2b–d). The characteristic absorption band of the quaternized pyridinium group at 1638 cm^−1^ is observed in polymer composite spectra (Figure 2b–d), beside the bands attributed to the polycation at 2800–3000 cm^−1^, 1516 cm^−1^, and 848 cm^−1^, which is similar to the observation reported by Marcilla et al. in the synthesis of polycations using ionic liquids [34,35]. These data confirm that the Fe_3_O_4_ magnetic nanoparticles are successfully imbedded into the QVP/AAm networks without oxidation into other iron oxides.

The XRD difffractograms of Fe_3_O_4_ and Fe_3_O_4_-QVP/AAm (Figure 3a,b) have the same diffraction patterns, which proves that the pure magnetite NPs are coated with QVP/AAm without the formation of any polymer composite. This result is supported by the appearance of the characteristic peaks and relative intensities, which match well with the characteristic of a standard Fe_3_O_4_ cubic crystalline sample [JCPDS file no. 19-0629, Figure 3a). The presence of the broad peak at 2-theta from 20° to 30° in the crystalline structure of QVP/AAm-Fe_3_O_4_ (Figure 3c) is consistent with an amorphous QVP/AAm phase that is indicated by the QVP/AAm diffractogram (Figure 3d). These data confirm that the magnetite NPs interact and are encapsulated with the PIL matrices. Scherrer’s formula is used to calculate the mean particle size (*D*_S_) of Fe_3_O_4_ nanocrystallites as *D*_S_ = *Kλ*/*β*cos*θ*, where *θ*, *β*, and *K* denote the position of the most intense peak with crystalline dimensions (311), the peak half-width, and Scherrer’s constant, respectively. The values of *D*_S_ for Fe_3_O_4_, Fe_3_O_4_-QVP/AAm, and QVP/AAm-Fe_3_O_4_ are 8.4, 20.5, and 45.1 ± 0.7 nm, respectively.

The surface morphologies of Fe_3_O_4_, Fe_3_O_4_-QVP/AAm, QVP/AAm-Fe_3_O_4_, and QVP/AAm can be estimated from the SEM and HR-TEM micrographs, as presented in Figure 4a–c and Figure 5a–d, respectively. The SEM images (Figure 4a–c) show the formation of non-uniform and stretched microspheres with QVP/AAm (Figure 4a) that are converted into uniform and dispersed nanospheres for the QVP/AAm-Fe_3_O_4_ composite (Figure 4b) and agglomerated nanospheres for Fe_3_O_4_-QVP/AAm (Figure 4c). These data elucidate show that the formation of the QVP monomer tends to copolymerize with AAm to form microgels. The presence of magnetite during the cross-linking copolymerization inhibits the growth of microgels to form nanogels. Meanwhile, it can be observed that the QVP/AAm microspheres are highly uniform and reduced in size when they interact with Fe_3_O_4_ using ultrasonication. 

It is obvious from the TEM images that the cubic Fe_3_O_4_-NPs (Figure 5a) convert into the ordered “core/shell” structure of QVP/AAm-Fe_3_O_4_ (Figure 5c) upon interaction with the microspheres of QVP/AAm (Figure 5b). An ordered Fe_3_O_4_ core with a mean diameter of approximately 5–10 nm is encapsulated inside mesoporous cross-linked PIL based on shell QVP/AAm approximately 200–400 nm to produce dispersed QVP/AAm-Fe_3_O_4_ composite (Figure 5c), while the presence of Fe_3_O_4_ during the crosslinking copolymerization of QVP/AAm forms agglomerates based on interacting core/shell magnetite NPs having a Fe_3_O_4_ core with thickness of 8–10 nm and QVP/AAm shell of 12–15 nm in thickness can be clearly observed (Figure 5d). Remarkably, because of their extraordinary perpendicular orientation, the mesoporous channels of the QVP/AAm-Fe_3_O_4_ composite nanospheres are easily accessible, preferring the adsorption and desorption of significant guest objects initiated by outer stimulants [36]. 

The thermal stabilities of the Fe_3_O_4_, Fe_3_O_4_-QVP/AAm, QVP/AAm-Fe_3_O_4_, and QVP/AAm samples and the magnetite contents of the PIL composites can be investigated using the TGA thermograms presented in Figure 6. It is observed that all the samples exhibit weight losses within or below 200 °C, which is attributed to the loss of adsorbed water molecules and hydroxyl groups from the Fe_3_O_4_ surfaces. The weight loss ranges from 6 to 10%, and it is increased for the QVP/AAm PIL. The QVP/AAm is the most stable polymer among other magnetite composites, which degrade before 300 °C (Figure 6). From the value of weight loss above 610 °C, the loading of the organic group bound to Fe_3_O_4_ can be calculated. The data confirmed that Fe_3_O_4_-QVP/AAm and QVP/AAm-Fe_3_O_4_ have magnetite content of 50 wt% and 40 wt%, respectively. It was also noticed that the weights of Fe_3_O_4_-QVP/AAm and QVP/AAm-Fe_3_O_4_ markedly decreased above 700 °C, and the organic QVP/AAm components on the Fe_3_O_4_ were decomposed completely up to 800 °C [37]. The weight loss around 650 °C corresponded to PIL chains that are directly grafted to the surface of the magnetic nanoparticles, either with Fe_3_O_4_-QVP/AAm or with QVP/AAm-Fe_3_O_4_. The data indicate that the QVP/AAm graft contents of Fe_3_O_4_-QVP/AAm and QVP/AAm-Fe_3_O_4_ are 35 wt% and 22 wt%, respectively, to confirm the proposed Figure 1 and Figure 2. 

The particle sizes and the surface charges of Fe_3_O_4_, Fe_3_O_4_-QVP/AAm, QVP/AAm-Fe_3_O_4_, and QVP/AAm samples were measured using dynamic light scattering (DLS) and zeta potential measurements as represented in Figure 7a–d. The data displayed in Figure 7a show that the dynamic sizes of Fe_3_O_4_, QVP/AAm-Fe_3_O_4_, and Fe_3_O_4_-QVP/AAm are 35.3, 80.9, and cluster aggregates (50 nm, 100 nm, and 10 µm), respectively. These values are much larger than those determined from the TEM images (Figure 4a–d). These refer to the magnetostatic (magnetic dipole-dipole) interactions of the magnetic particles without any external magnetic field. These interactions lead to the formation of closed rings and long open loops of magnetic particles without preferential spatial orientation [33]. The polydispersity index (PDI) values determined from the DLS data of Fe_3_O_4_, Fe_3_O_4_-QVP/AAm, QVP/AAm-Fe_3_O_4_, and QVP/AAm are 0.232, 0.732, 0.510, and 0.310, respectively. These data confirm the encapsulation of magnetite with QVP/AAm (Figure 7d) modifies the dispersity of QVP/AAm more than its incorporation during the crosslinking and copolymerization of QVP/AAm (Figure 7c). The surface charges (zeta potentials) of Fe_3_O_4_, Fe_3_O_4_-QVP/AAm, QVP/AAm-Fe_3_O_4_, and QVP/AAm are −15.30 mV, 28.90 mV, 40.81 mV, and 17.82 mV, respectively. The increase in the surface charges of the particles beyond 25 mV indicates the good dispersion of the prepared microparticles and nanoparticles in water [38]. Moreover, the positive charges on the surfaces of Fe_3_O_4_-QVP/AAm, QVP/AAm-Fe_3_O_4_, and QVP/AAm indicate their capability to interact with the negative charge on the surfaces of the pollutants such as anionic dyes. 

### 3.2. Application of Magnetic QVP/AAm Composite for CR Adsorbents

The magnetic characteristics of Fe_3_O_4_, Fe_3_O_4_-QVP/AAm, and QVP/AAm-Fe_3_O_4_ were determined using a vibrating sample magnetometer (VSM) at room temperature. Their magnetic hysteresis loops are presented in Figure 8a–c. The saturation magnetizations of Fe_3_O_4_, Fe_3_O_4_-QVP/AAm, and QVP/AAm-Fe_3_O_4_ are 75.29 emu·g^−1^, 45.64 emu·g^−1^, and 64.56 emu·g^−1^, respectively. These data show that the magnetization of magnetite is reduced after functionalization with non-magnetic PILs owing to their shielding effect [39,40]. The higher magnetite content of Fe_3_O_4_-QVP/AAm leads to higher magnetization than QVP/AAm-Fe_3_O_4_ as illustrated in the TGA analysis (Figure 6) [40]. The values of the saturation magnetization indicate that the prepared magnetite composites are superparamagnetic materials [40]. In this way, the prepared materials are used to remove the anionic CR dye as a pollutant adsorbate of water. CR dye is a derivative of benzidine and napthoic acidic azo dye. It is used in the textile industry, and it decomposes in industrial wastewater to form carcinogenic products [41]. Moreover, it is a skin, eye, and gastrointestinal irritant. Further, it may affect blood factors such as clotting and induce somnolence and respiratory problems [42,43]. In this section, we investigate the optimum conditions (such as adsorbent content, CR dye concentration, contact time, pH, ionic strength, and temperature of water) required to remove CR dye from water using the prepared magnetic composites based on cross-linked PIL.

The pore structure and surface area of Fe_3_O_4_-QVP/AAm, QVP/AAm-Fe_3_O_4_ are determined from the nitrogen adsorption-desorption isotherms at 77 K to determine their porosity as porous adsorbents and are presented in Figure 9. The humidity was removed from samples pores by pretreating the sample at a temperature of 353 K. The samples were also heated under vacuum up to 423 K before measuring their surface area and pore sizes. The sorption isotherms of Fe_3_O_4_-QVP/AAm, QVP/AAm-Fe_3_O_4_ (Figure 8) gave rise to type I. The BET surface area (S_BET_; m^2^·g^−1^), pore size diameters (*D*; nm), and pore volume (V_total_; cm^3^·g^−1^) of Fe_3_O_4_-QVP/AAm are 64, 15.56 and 0.1169, respectively. The *D*, S_BET_ and V_total_ values of QVP/AAm-Fe_3_O_4_ are 115 m^2^·g^−1^, 24.63 nm, and 0.1929 cm^3^·g^−1^, respectively. The data elucidate that the pore sizes increased in case of QVP/AAm-Fe_3_O_4_ due to linking of magnetite with silica which activates the silica surfaces and increases their surface area. These data confirm the formation of the porous structures of Fe_3_O_4_-QVP/AAm, QVP/AAm-Fe_3_O_4_ as confirmed from TEM (Figure 4) to enhance their application as an adsorbent.

The prepared Fe_3_O_4_, Fe_3_O_4_-QVP/AAm, QVP/AAm-Fe_3_O_4_, and QVP/AAm are highly dispersed in the different pH aqueous solutions. QVP/AAm does not separate easily from its aqueous solution, especially in the presence of the CR dye, which cannot be estimated accurately by UV spectra even after filtration. Moreover, Fe_3_O_4_ and Fe_3_O_4_-QVP/AAm achieved low removal efficiencies even at low CR concentrations with longer contact times, up to 48 h. The QVP/AAm-Fe_3_O_4_ is selected as the most effective adsorbent among the prepared materials based on PILs as compared to other PILs materials presented in the literature [17,44,45,46,47]. Gharehbaghi and Shemirani [44] used a new technique based on solvent extraction of Congo red from wastewater using the ionic liquid 1-hexyl-3-methylimidazolium bis(trifluormethylsulfonyl)imide. 

The effect of contact time between the QVP/AAm-Fe_3_O_4_ adsorbent (at a concentration of 150 mg·L^−1^) and CR dye in water (0.35 mmol·L^−1^) is presented in Figure 10 to assess its ability to adsorb fast. The data presented in Figure 8 show that QVP/AAm-Fe_3_O_4_ achieves its equilibrium adsorption capacity (*q*_e_) after a contact time (*t*) of 35 min at room temperature. It achieves an adsorption capacity of 903 mg·g^−1^ in a short time compared to hyperbranched PILs and magnetic composites [17,18], which achieved 500 mg·g^−1^ after 10 h. Therefore, in the adsorption studies described in the following discussion, a contact time of 35 min is selected to measure the other optimum conditions.

The impact of QVP/AAm-Fe_3_O_4_ adsorbent dosage on *q*_e_ and *E* (in%) is shown in Figure 11. The data show that the QVP/AAm-Fe_3_O_4_ achieves the maximum removal efficiency (100%) and *q*_e_ at 850 mg·g^−1^ using an adsorbate concentration of 150 mg·L^−1^, which is lower than that obtained in the literature ranging between 370 mg·L^−1^ and 600 mg·L^−1^ to remove 92.5% of the CR dye [17,18]. However, the *q*_e_ value declined with the increasing amount of QVP/AAm-Fe_3_O_4_ adsorbent. The effect of the initial CR dye concentration on the value of *E* (%) for the QVP/AAm-Fe_3_O_4_ adsorbent is investigated at room temperature as presented in Figure 12. Different initial concentrations of CR dye, i.e., *C*_0_ values, ranging from 0.05 to 0.25 mmol·L^−1^ are used (Figure 11). As seen in Figure 11, at the *C*_0_ values of 0.2 mmol·L^−1^ for CR dye, the *E* (%) and *q*_e_ values are 98.9% and 940 mg·g^−1^, respectively. The optimum initial CR dye concentration is 0.2 mmol·L^−1^, which is suitable for the fast removal of CR dye from water. The maximum removal efficiency and *q*_e_ values in the present work are much higher than those of the formerly reported adsorbents for CR dye [17,18,44,45,46,47].

The impact of pH of CR dye at an initial concentration of 0.2 mmol·L^−1^ on the value of E (%) for QVP/AAm-Fe_3_O_4_ at a concentration of 150 mg·L^−1^ for contact time of 35 min is presented in Figure 13. The data confirm that the optimum water neutral pH of 7 is suitable for achieving the maximum adsorption removal efficiencies that are reduced at acidic pH < 7 and stable in alkaline pH > 7. The decreasing pH values at the active sites of QVP/AAm-Fe_3_O_4_ could be caused by the aggregation of magnetic composites at acidic pH values [48]. The zeta potential of QVP/AAm-Fe_3_O_4_ is less positive at a pH of 2; it then increases from 6.30 mV to 40.81 mV as the pH value ranges from 2 to 7. It is widely known that the electrostatic attraction between the cationic charged adsorbent and the negative dye molecules increases under acidic conditions [49]. Consequently, the aggregation of the Fe_3_O_4_ composite and the protonation of the amino groups of CR dye reduce the negative charges on CR dye and affect the electrostatic attractions as represented in Figure 3. The encapsulation of magnetite into the QVP/AAm protects it from damage in the strong acid solution [50]. Therefore, an initial pH of 7 is selected for further experiments, to determine the uptake of the CR dye. The dye solution in the textile dyeing process usually contains NaCl, which promotes the dye adsorption of textile fibers. [51]. Figure 13 represents the effect of the presence of 4000 mg·L^−1^ of NaCl and acidic pH of the CR solution on the value of E (in%) on the CR dye at an initial concentration 0.2 mmol·L^−1^ in the presence of QVP/AAm-Fe_3_O_4_ at a concentration of 150 mg·L^−1^ for a contact time of 35 min. As seen from Figure 11, the presence of NaCl increases the CR dye adsorption, as magnetite composite dispersion increases in seawater [52]. 

### 3.3. Adsorption Kinetics Isotherms and Mechanism of QVP/AAm-Fe_3_O_4_

The interaction mechanism between the surface of the prepared QVP/AAm-Fe_3_O_4_ and the CR dye molecules, beside the diffusion of CR molecules inside the adsorbent pores, can be investigated from the adsorption kinetics and isotherms. The homogeneity and heterogeneity of the QVP/AAm-Fe_3_O_4_ surfaces can be investigated using the Langmuir and Freundlich adsorption models Equations:

(*C*_e_/*Q*_e_) = [(1/*Q*_max_*K*_l_) + (*C*_e_/*Q*_max_)
(3)


log(*Q*_e_) = log(*K*_f_) + [(1/*n*) log(*C*_e_)]
(4)


The constants *n* (in g·L^−1^), *K*_l_ (in L·mg^−1^), and *K*_f_ [in (mg·g^−1^)(L·mg^−1^)^(1/*n*)^] are the empirical constant, Langmuir constant, and Freundlich constant, respectively. *Q*_e_ and *Q*_max_ (in mg·g^−1^) are the equilibrium and maximum amounts of CR adsorbate, respectively. *C*_e_ (mg·L^−1^) is the concentration of CR dye in the aqueous solution at equilibrium. Equations (3) and (4) should obey a linear relation with the highest linear coefficient (*R*^2^). The adsorption parameters of the Langmuir and Freundlich equations are listed in Table 1. The data listed in Table 1 confirm that the adsorption of the CR molecules obeys the Langmuir adsorption isotherm more than the Freundlich model. This elucidates the homogeneity of QVP/AAm-Fe_3_O_4_ with the formation of the CR monolayer onto the composite surface. These data show that the dispersion of magnetite onto the QVP/AAm-Fe_3_O_4_ surfaces facilitates the formation of the CR monolayer on the adsorbent surfaces (Figure 3). 

The rate and CR adsorption mechanism of QVP/AAm-Fe_3_O_4_ can be estimated and analyzed by using pseudo-first-order and pseudo-second-order models [53] as represented in Figure 14a,b, respectively and summarized in Table 2. The higher correlation coefficient (*R*^2^) and the calculated adsorption capacity (q_calc._; mg·g^−1^) values of the fitted curves obtained by pseudo-second-order kinetic models with the experimental adsorption capacity (q_exp._; mg·g^−1^) value confirm that the pseudo-second-order model can describe the adsorption process better with a higher and faster adsorption rate (*K*_2_; Table 2) as compared to other PILs [17,18,44,45,46,47]. These data show that the strong hydrogen bond, ion exchange CR anions, and Cl^−^ counter-ions, and the electrostatic interaction between the QVP/AAm-Fe_3_O_4_ adsorbent (Figure 3) are responsible for the chemisorption nature of QVP/AAm-Fe_3_O_4_ towards CR dye [54,55]. 

Thermodynamic parameters such as standard Gibbs energy (∆*G*_o_; in kJ·mol^−1^), enthalpy (∆*H*_o_; in kJ·mol^−1^), and entropy (∆*S*_o_; in J·mol^−1^·K) were calculated using the following equations:

∆*G*_o_ = −*RT* ln(*C*_e_*A*/*C*_e_)
(5)


log(*C*_e_*A*/*C*_e_) = ∆*S*_o_/2.303*R* − ∆*H*_o_/2.303*RT*
(6)

where *C*_e_*A*, *R*, and *T* are the adsorbent concentration (in mg·L^−1^), gas constant (8.314 J mol^−1^·K^−1^), and the aqueous solution temperature (in K), respectively. The values of ∆*G*_o_, ∆*H*_o_ and ∆*S*_o_ are summarized in Table 3. The equilibrium concentration constant (*K*_c_) can be calculated from the relation (*C*_e_*A*/*C*_e_) and plotted in Figure 15. As can be seen from Table 3, the negative value of ∆*G*_o_ indicates the spontaneous nature of the CR dye adsorption by the QVP/AAm-Fe_3_O_4_ adsorbent. The negative value of ∆*H*_o_ (more than −20 kJ·mol^−1^) confirms the chemical adsorption and exothermic nature of the adsorption process [56]. Moreover, this value is in agreement with the results of the experiment, which confirms the increase in the adsorption capacity at lower temperatures. This also confirms that the negative value of ∆*S* corresponds to a decrease in the degrees of freedom of the CR species with increasing temperature. Finally, the mechanism of the adsorption of CR dye on the QVP/AAm-Fe_3_O_4_ surfaces can be proposed (Figure 2). The CR dye was first adsorbed into the QVP/AAm-Fe_3_O_4_ polymer matrix in the adsorption process. The dye began to fill the pores with increasing adsorption capacity by ion exchange with the chemical binding owing to lower particle sizes of magnetite, uniform dispersion, and formation of pores.

The adsorption efficiency of QVP/AAm-Fe_3_O_4_ nanocomposite was compared with other sorbents as shown in Table 4. The present adsorbent shows higher adsorption capacity in a shorter time when comparing with other adsorbents [57,58,59,60,61]. QVP/AAm-Fe_3_O_4_ nanocomposite exhibited high adsorption capacity because of the porous framework and the presence of quaternary ammonium salt in the polymer backbone that interacts with the CR dye.

### 3.4. The Reusability of QVP/AAm-Fe_3_O_4_ Nanocomposite

The desorption-sorption experiments were carried out in a basic aqueous solution as described in the experimental part. Table 5 displays the removal efficiency of QVP/AAm-Fe_3_O_4_ nanocomposite in different cycles. The desorption of CR occurred in basic medium not in the neutral one which means that there is strong ion-ion interaction between CR dye and the nanocomposite [62]. The magnetite nanoparticles are strongly capped and protected with QVP/AAm so that there is no leaching of magnetite during six reusability cycles. There is a relatively stability of the CR removal efficiency during the six reuse cycles. It can be seen that the four cycles have similar CR dye desorption and adsorption results when compared with the first adsorption. The fifth cycle shows that the CR dye adsorption efficiencies of QVP/AAm-Fe_3_O_4_ polymer were decreased to 75%. This could be because of the destruction of the electrostatic attraction between the composite and CR molecules, whereas hydrogen bonding attraction and physical adsorption still exist after the treatment without the destruction of the magnetic characteristics of QVP/AAm-Fe_3_O_4_ [17,18,44,45,46,47].

## 4. Conclusions

A new superparamagnetic cross-linked mesoporous core/shell magnetic PIL was prepared based on 4-vinylpyridine (QVP)/AAm-Fe_3_O_4_ nanocomposite to remove anionic Congo red dye from industrial wastewaters. The QVP/AAm-Fe_3_O_4_ nanocomposite PIL possesses good magnetization and uniform, accessible mesopores. The nanocomposite exhibits fast and high adsorption performance and rapid adsorption capacity for CR dye in water. Moreover, using QVP/AAm-Fe_3_O_4_ nanocomposite as an adsorbent leads to fast adsorption and desorption kinetics, rapid magnetic separation process, and no secondary pollution in the adsorption process for CR dye from water. These characteristics of the QVP/AAm-Fe_3_O_4_ nanocomposite make it a highly efficient adsorbent for the removal of anionic acidic CR contaminants from industrial wastewater as compared to other composites described in the literature [17,18,44,45,46,47].

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
