# Peer review of "Synthesis of New Magnetic Crosslinked Poly (Ionic Liquid) Nanocomposites for Fast Congo Red Removal from Industrial Wastewater"

_nanomaterials, 2019, doi:10.3390/nano9091286_

Round 1
Reviewer 1 Report
The authors present the preparation of a polymer based on a cross-linked bis(4-vinylpyridinium) dicationic monomer functionalized with magnetic nanoparticles. The materials are characterized and investigated as sorbents for the removal of dye from wastewater.
The dicationic monomer used by the authors appears to be novel, so this study does thoroughly characterize a new material. However, there are a number of inconsistencies and omissions that need to be addressed.
First, there are no ionic liquids anywhere in this study. 4-vinylpyridine, di(2-chloroethyl)ether, and all of the cross linking agents are molecular. Fe3O4 is solid. The monomer, bis(2-(4-vinyl-N-pyridinium))ethyl ether dichloride, is never isolated or shown to be a liquid, nor would I expect this compound likely to have a low melting point. The materials themselves are isolated as solids, exclusively handled as solids, and never shown to be able to melt. Most references to ionic liquids in the introduction and comparisons to PILs in the results and discussion are irrelevant. Polymers based on 4-vinylpyridine and pyridinium are already well known and even commercially available as ion exchange resins. These are the materials which this study is relevant to.
The chemical characterization of the polymer is insufficient, given that it is a multi-step reaction. The IR signals the authors use as evidence that the reactions have gone forward are all for functional groups that are also present in the starting materials. At a minimum the authors should isolate and characterize the dipyridinium cation monomer or use chromatography to prove that the 5-hour reaction time is sufficient for the first part of the reaction to go to completion.
Some bulk properties are inferred from the TEM measurement, such as the presence of perpendicular channels, which are probably artifacts of slicing the sample for the measurement. For instance, the authors later note inconsistencies between the DLS and TEM measurements and attribute them to magnetic behavior, but it is more likely that the morphologies simply change on isolating, drying, and slicing the sample for TEM.
The authors attempt to model the sorption mechanism by monitoring sorption capacity with time. They are effectively monitoring the rate of product formation, which is the rate of the overall reaction (dye + sorbent -> dye-sorbent complex). In many sorbent studies this is treated as a pseudo-first order process because experiments are designed so that the change in concentration of the free sorbent sites is negligible, the the derivative of sorbent free sites with respect to time can be assumed to be zero, and therefore the change in concentration of the analyte can be assumed to be the reaction rate. However, this sorbent has an exceptionally high capacity for the sorbate, close to 1 g/1 g. In this case, the concentration of free sorbent and sorbate are both changing. The authors observe a second order rate law because the reaction is likely first order with respect to each reactant.
One of the most important experimental omissions is that the use of a magnetic technique to isolate the sorbent is never described. Although desorption and recycling experiments are described, I was unable to find where the isolation of the loaded sorbent was described (eg. centrifugation, filtering).
Author Response
Reviewer 1
Comments and Suggestions for Authors
First, there are no ionic liquids anywhere in this study. 4-vinylpyridine, di(2-chloroethyl)ether, and all of the cross linking agents are molecular. Fe3O4 is solid. The monomer, bis(2-(4-vinyl-N-pyridinium))ethyl ether dichloride, is never isolated or shown to be a liquid, nor would I expect this compound likely to have a low melting point. The materials themselves are isolated as solids, exclusively handled as solids, and never shown to be able to melt. Most references to ionic liquids in the introduction and comparisons to PILs in the results and discussion are irrelevant. Polymers based on 4-vinylpyridine and pyridinium are already well known and even commercially available as ion exchange resins. These are the materials which this study is relevant to.
Answer
Poly ionic liquid here refers to the quaternized pyridinium nitrogen (ammonium ionic liquid). The final composite won’t have melting point due to its crosslinking which cannot melt and solubilize in any organic solvent. there are many works related this subject one of them is N. Sahiner, A. O. Yaser and N. Aktas prepared Poly(4-vinylpyridine)-based polymeric ionic liquid by the polymerization of (4-VP) in the presence of methylene bis acrylamide as a cross linker followed by addition of di bromo alkane to quaternize the pyridinium nitrogen [N. Sahiner, A. O. Yaser and N. Aktas, international journal of hydrogen energy, Vol. 41, 20562-20572, 2016] .
The chemical characterization of the polymer is insufficient, given that it is a multi-step reaction. The IR signals the authors use as evidence that the reactions have gone forward are all for functional groups that are also present in the starting materials. At a minimum the authors should isolate and characterize the dipyridinium cation monomer or use chromatography to prove that the 5-hour reaction time is sufficient for the first part of the reaction to go to completion.
Answer
The formation of the final product as a solid crosslinked PIL is an evidence for the complete formation of the dipyridinium cation monomer. The reaction lasted for 5h at room temperature and then another 5 h at 55 0C and this time is sufficient to complete the first step of the reaction. N. Bicak and M. Gazi quaternized poly vinyl pyridine by 2-Chloroacetamide in DMF solvent at room temperature for 1h [N. Bicak and M. Gazi Journal of macromolecular science Vol. A40, 585-591, 2003].
Some bulk properties are inferred from the TEM measurement, such as the presence of perpendicular channels, which are probably artifacts of slicing the sample for the measurement. For instance, the authors later note inconsistencies between the DLS and TEM measurements and attribute them to magnetic behavior, but it is more likely that the morphologies simply change on isolating, drying, and slicing the sample for TEM.
Answer
Different than other nanoparticles that may give the same value of particle size determined by TEM or DLS, Iron oxide nanoparticles tend to aggregate when dispersed into solvents. Magnetite nanoparticles exhibit a tendency to agglomerate in order to reduce their surface energy by strong magnetic dipole-dipole interactions between particles [Surowiec, Z.; Budzyński, M.; Durak, K.; Czernel, G. Synthesis and characterization of iron oxide magnetic nanoparticles. Nukleonika 2017, 62, 73-77].
BET measurement added to the text to determine the porosity of composites
The authors attempt to model the sorption mechanism by monitoring sorption capacity with time. They are effectively monitoring the rate of product formation, which is the rate of the overall reaction (dye + sorbent -> dye-sorbent complex). In many sorbent studies this is treated as a pseudo-first order process because experiments are designed so that the change in concentration of the free sorbent sites is negligible, the derivative of sorbent free sites with respect to time can be assumed to be zero, and therefore the change in concentration of the analyte can be assumed to be the reaction rate. However, this sorbent has an exceptionally high capacity for the sorbate, close to 1 g/1 g. In this case, the concentration of free sorbent and sorbate are both changing. The authors observe a second order rate law because the reaction is likely first order with respect to each reactant.
Answer: The data of the present work treated with standard kinetic modules as discussed in all data reported in the literature. The higher correlation coefficient (R2) and the calculated qe,cal values of the fitted curves obtained by pseudo-second-order kinetic models with the experimental qe,exp value confirm that the pseudo-second-order model can describe the adsorption process better with a higher and faster adsorption rate (K2; Table 2) as compared to other PILs [Cheng, J.; Shi, L.; Lu, J. Amino ionic liquids-modified magnetic core/shell nanocomposite as an efficient adsorbent for dye removal. Journal of Industrial and Engineering Chemistry 2016, 36, 206-214].
One of the most important experimental omissions is that the use of a magnetic technique to isolate the sorbent is never described. Although desorption and recycling experiments are described, I was unable to find where the isolation of the loaded sorbent was described (eg. centrifugation, filtering).
Answer
It is well known that the magnetic materials will be isolated with an external magnet and this is the main reason to prepare magnetic materials. The sentence “The usage of external magnetic field to isolate the sorbent” was added in the experimental part.

Reviewer 2 Report
This manuscript reports the synthesis and characterization of magnetic cross-linked PIL and 4-vinylpyridine-co-acrylamide (QVP/AAm) composite for adsorptive removal of Congo red dye from waste water. The work is interesting and timely. Manuscript can be accepted for publication after considering the following comments:
Line “To prepare polymerizable quaternized 4-vinylpyridine (QVP), VP (6.5 g, 0.06 mol) and DCDE (4.25 g, 0.03 mol) were dissolved in 25 mL of DMF” is confusing. Visibility of Figure 1 should be increases Surface area are quite significant for adsorption. BET analysis of synthesized materials should be provided. Authors should add recyclability of absorbent at least up to 5 cycles. Stability of adsorbent are quite important, characterization of adsorbent before and after adsorption experiments should be added. For the sake of completeness in order to attract more attention and enhance the readership of the scientific communities on adsorption, the recent related references are suggested-ACS Sustainable Chem. Eng.2019, 7, 4, 3772-3782. ACS Appl. Mater. Interfaces 2019, 11, 20, 18165-18177. New Journal of Chemistry, DOI: 10.1039/C9NJ02344E.
A comparative table with other ionic liquid based materials should be included.
Author Response
Reviewer 2:
Comments and Suggestions for Authors
The comments are:
Line “To prepare polymerizable quaternized 4-vinylpyridine (QVP), VP (6.5 g, 0.06 mol) and DCDE (4.25 g, 0.03 mol) were dissolved in 25 mL of DMF” is confusing.
Answer:
Line “To prepare polymerizable quaternized 4-vinylpyridine (QVP), VP (6.5 g, 0.06 mol) and DCDE (4.25 g, 0.03 mol) were dissolved in 25 mL of DMF has been modified to
“To quaternize 4-vinylpyridine (QVP), 4-vinylpyridine (6.5 g, 0.06 mol) and dichlorodiethylether (4.25 g, 0.03 mol) were dissolved in 25 mL of DMF”
Visibility of Figure 1 should be increases
Answer
Resolution of Figure 1 has been enhanced.
Surface area are quite significant for adsorption. BET analysis of synthesized materials should be provided.
Answer:
New BET measurements added in the text.
Authors should add recyclability of absorbent at least up to 5 cycles.
Answer: new data added for 5 cycles
Stability of adsorbent are quite important, characterization of adsorbent before and after adsorption experiments should be added.
For the sake of completeness in order to attract more attention and enhance the readership of the scientific communities on adsorption, the recent related references are suggested-
ACS Sustainable Chem. Eng.2019, 7, 4, 3772-3782. ACS Appl. Mater. Interfaces 2019, 11, 20, 18165-18177. New Journal of Chemistry, DOI: 10.1039/C9NJ02344E.
A comparative table with other ionic liquid based materials should be included.
Answer:
A comparative table with other ionic liquid based materials has been added to the manuscript.
ACS Sustainable Chem. Eng.2019, 7, 4, 3772-3782. ACS Appl. Mater. Interfaces 2019, 11, 20, 18165-18177. New Journal of Chemistry, DOI: 10.1039/C9NJ02344E reference has been added (reference 2)

Round 2
Reviewer 1 Report
The authors have not made revisions to the text based on most of my comments, and my opinion about this manuscript remains unchanged. Below are my rebuttals to the authors' responses, in order to clarify my position and aid them in making revisions.
Author Response: Poly ionic liquid here refers to the quaternized pyridinium nitrogen (ammonium ionic liquid). The final composite won’t have melting point due to its crosslinking which cannot melt and solubilize in any organic solvent. there are many works related this subject one of them is N. Sahiner, A. O. Yaser and N. Aktas prepared Poly(4-vinylpyridine)-based polymeric ionic liquid by the polymerization of (4-VP) in the presence of methylene bis acrylamide as a cross linker followed by addition of di bromo alkane to quaternize the pyridinium nitrogen [N. Sahiner, A. O. Yaser and N. Aktas, international journal of hydrogen energy, Vol. 41, 20562-20572, 2016]
Rebuttal: Ionic liquids are not defined by the presence of a certain functional group. For instance, all alkylpyridinium salts are not considered ionic liquids. Ionic liquids are generally defined by melting point (see Hallet and Welton, Chem. Rev. Cross linked poly-(4-vinylpyridine)resins are commercially used as ion exchange resins, and I maintain that this is the class of materials needed for comparison (not ionic liquids).
Author Response: The formation of the final product as a solid crosslinked PIL is an evidence for the complete formation of the dipyridinium cation monomer. The reaction lasted for 5h at room temperature and then another 5 h at 55 0C and this time is sufficient to complete the first step of the reaction. N. Bicak and M. Gazi quaternized poly vinyl pyridine by 2-Chloroacetamide in DMF solvent at room temperature for 1h [N. Bicak and M. Gazi Journal of macromolecular science Vol. A40, 585-591, 2003].
Rebuttal: It is on the burden of the authors to prove that "The formation of the final product as a solid crosslinked PIL" occurred. This cannot be used by itself as evidence that the reaction happened, because the cross-links themselves are not directly observable. They must be detected by spectroscopic or structural means. The fact that a similar reaction has been reported before is also not sufficient evidence; the authors have changed the procedure in order to make a novel material. There is the possibility that these changes affect the reaction rate or cause side products to form that were not observed in the prior report.
I do note that if the authors have no means of characterizing the material beyond what they have provided, this does not mean the material does not work. It simply means the authors can only hypothesize about its structure based on the chemical reaction attempted and cannot claim to know it for sure.
Author Response: Different than other nanoparticles that may give the same value of particle size determined by TEM or DLS, Iron oxide nanoparticles tend to aggregate when dispersed into solvents. Magnetite nanoparticles exhibit a tendency to agglomerate in order to reduce their surface energy by strong magnetic dipole-dipole interactions between particles [Surowiec, Z.; Budzyński, M.; Durak, K.; Czernel, G. Synthesis and characterization of iron oxide magnetic nanoparticles. Nukleonika 2017, 62, 73-77].
Rebuttal: I am satisfied with the additional experiment added here.
Author Response: The data of the present work treated with standard kinetic modules as discussed in all data reported in the literature. The higher correlation coefficient (R2) and the calculated qe,cal values of the fitted curves obtained by pseudo-second-order kinetic models with the experimental qe,exp value confirm that the pseudo-second-order model can describe the adsorption process better with a higher and faster adsorption rate (K2; Table 2) as compared to other PILs [Cheng, J.; Shi, L.; Lu, J. Amino ionic liquids-modified magnetic core/shell nanocomposite as an efficient adsorbent for dye removal. Journal of Industrial and Engineering Chemistry 2016, 36, 206-214].
Rebuttal: The model I am proposing is also a standard kinetic model (see, for instance, Petrucci and Hardwood, General Chemistry: Principles and Modern Applications, Macmillan: New York, 1993, pp. 514-536). A chemical reaction (including the formation of a sorbent-solute complex) can be described with the following rate law: R = k[A]m[B]n... where [A], [B], etc. are concentrations of reactants and k is the rate constant. The sum of the exponents m, n, etc. give the order of the reaction (first order, second order, etc.). The authors observe a second order rate for the reaction and propose that the reaction is pseudo second-order with respect to the concentration of Congo red; I believe it is more likely the reaction is first order with respect to both the sorbent and the dye. The overall rate of the reaction, which is what would be measured by this experiment, would be second-order (because m and n are both equal to 1, so m+n = 2).
Author Response: It is well known that the magnetic materials will be isolated with an external magnet and this is the main reason to prepare magnetic materials. The sentence “The usage of external magnetic field to isolate the sorbent” was added in the experimental part.
Rebuttal: The burden is on the authors to demonstrate and prove that the material has this expected magnetic behavior, therefore showing that the material works. The application of the external magnetic field needs to be described more specifically. Was a magnet put in the suspension and recovered? Was a magnetic field applied to settle the suspension so that the solution could be decanted?
Author Response
Reviewer 1
Rebuttal: Ionic liquids are not defined by the presence of a certain functional group. For instance, all alkylpyridinium salts are not considered ionic liquids. Ionic liquids are generally defined by melting point (see Hallet and Welton, Chem. Rev. Cross linked poly-(4-vinylpyridine)resins are commercially used as ion exchange resins, and I maintain that this is the class of materials needed for comparison (not ionic liquids).
Answer: The crosslinked PIL can be obtained from the crosslinking of polymerizable IL monomers with crosslinkers containing at least two double bonds by radical polymerization technique. In this respect, the present work prepared and purified liquid IL crosslinker QVP as reported in the experimental section. The chemical structure of the QVP elucidated by 1HNMR and 13CNMR spectra represented in Figure 1a and b, respectively. Also the QVP is liquid as reported in the experimental section
Rebuttal: It is on the burden of the authors to prove that "The formation of the final product as a solid crosslinked PIL" occurred. This cannot be used by itself as evidence that the reaction happened, because the cross-links themselves are not directly observable. They must be detected by spectroscopic or structural means. The fact that a similar reaction has been reported before is also not sufficient evidence; the authors have changed the procedure in order to make a novel material. There is the possibility that these changes affect the reaction rate or cause side products to form that were not observed in the prior report.
I do note that if the authors have no means of characterizing the material beyond what they have provided, this does not mean the material does not work. It simply means the authors can only hypothesize about its structure based on the chemical reaction attempted and cannot claim to know it for sure.
Answer: QVP separated and characterized
Rebuttal: The model I am proposing is also a standard kinetic model (see, for instance, Petrucci and Hardwood, General Chemistry: Principles and Modern Applications, Macmillan: New York, 1993, pp. 514-536). A chemical reaction (including the formation of a sorbent-solute complex) can be described with the following rate law: R = k[A]m[B]n... where [A], [B], etc. are concentrations of reactants and k is the rate constant. The sum of the exponents m, n, etc. give the order of the reaction (first order, second order, etc.). The authors observe a second order rate for the reaction and propose that the reaction is pseudo second-order with respect to the concentration of Congo red; I believe it is more likely the reaction is first order with respect to both the sorbent and the dye. The overall rate of the reaction, which is what would be measured by this experiment, would be second-order (because m and n are both equal to 1, so m+n = 2).
Answer: The kinetic model used in this work have been used in the almost of articles studied the adsorbent to remove the water inorganic or organic pollutants as recommended models.
Rebuttal: The burden is on the authors to demonstrate and prove that the material has this expected magnetic behavior, therefore showing that the material works. The application of the external magnetic field needs to be described more specifically. Was a magnet put in the suspension and recovered? Was a magnetic field applied to settle the suspension so that the solution could be decanted?
Answer: I added sentences marked with the red colour in the experimental section top clarify the magnetic separation of the adsorbent beside their magnetic properties that measured and evaluated to confirm the superior magnetic properties of the materials
Reviewer 2 Report
All the references suggested by the reviewers should be included
Author Response
Reviewer 2:
Comments and Suggestions for Authors The comments are:
1- Line “To prepare polymerizable quaternized 4-vinylpyridine (QVP), VP (6.5 g, 0.06 mol) and DCDE (4.25 g, 0.03 mol) were dissolved in 25 mL of DMF” is confusing.
Answer: Line “To prepare polymerizable quaternized 4-vinylpyridine (QVP), VP (6.5 g, 0.06 mol) and DCDE (4.25 g, 0.03 mol) were dissolved in 25 mL of DMF has been modified to “To quaternize 4-vinylpyridine (QVP), 4-vinylpyridine (6.5 g, 0.06 mol) and dichlorodiethylether (4.25 g, 0.03 mol) were dissolved in 25 mL of DMF”
2- Visibility of Figure 1 should be increases
Answer Resolution of Figure 1 has been enhanced.
3- Surface area are quite significant for adsorption. BET analysis of synthesized materials should be provided.
Answer: New BET measurements added in the text.
4- Authors should add recyclability of absorbent at least up to 5 cycles. Answer: new data added for 5 cycles Stability of adsorbent are quite important, characterization of adsorbent before and after adsorption experiments should be added.
5- For the sake of completeness in order to attract more attention and enhance the readership of the scientific communities on adsorption, the recent related references are suggested-
ACS Sustainable Chem. Eng.2019, 7, 4, 3772-3782. ACS Appl. Mater. Interfaces 2019, 11, 20, 18165-18177. New Journal of Chemistry, DOI: 10.1039/C9NJ02344E.
Answer: A comparative table with other ionic liquid based materials should be included. Answer: A comparative table with other ionic liquid based materials has been added to the manuscript. ACS Sustainable Chem. Eng.2019, 7, 4, 3772-3782. ACS Appl. Mater. Interfaces 2019, 11, 20, 18165-18177. New Journal of Chemistry, DOI: 10.1039/C9NJ02344E reference has been added (reference 2
Round 3
Reviewer 1 Report
The authors have, importantly, now isolated the monomer and presented evidence that it is an ionic liquid. In this sense, the authors have shown that their system is a polymerized ionic liquid analagous to poly (1-vinyl-3-methyl)imidazolium salts and others accepted by the field.
I encourage the authors to consider their system critically from a molecular standpoint rather than be only comparative with the literature, but since the kinetic data and explanation provided by the authors does have literature precedence I will not force this issue.
The only revision I recommend is improved graphics for Fig. 1 and Scheme 3. I do not need to review this manuscript again.
Author Response
I did the minor revision and I increased the resolution of scheme 3 and figure 1